# Epigenetic Integrity of Orthodox Seeds Stored under Conventional and Cryogenic Conditions

**Beata P. Plitta-Michalak** [1,2,*], **Mirosława Z. Naskręt-Barciszewska** [3], **Jan Barciszewski** [3,4], **Paweł Chmielarz** [2] **and Marcin Michalak** [1,2,*]

1. Department of Plant Physiology, Genetics and Biotechnology, University of Warmia and Mazury in Olsztyn, M Oczapowskiego 1A, 10-721 Olsztyn, Poland
2. Institute of Dendrology, Polish Academy of Sciences, Parkowa 5, 62-035 Kórnik, Poland; pach@man.poznan.pl
3. Institute of Bioorganic Chemistry, Polish Academy of Sciences, Z. Noskowskiego 12/14, 61-704 Poznań, Poland; miroslawa.barciszewska@ibch.poznan.pl (M.Z.N.-B.); Jan.Barciszewski@ibch.poznan.pl (J.B.)
4. NanoBioMedical Centre, Adam Mickiewicz University, Wszechnicy Piastowskiej 3, 61-614 Poznań, Poland
* Correspondence: beata.plitta-michalak@uwm.edu.pl (B.P.P.-M.); m.michalak@uwm.edu.pl (M.M.)

**Abstract:** The level of 5-methylcytosine ($m^5C$) in DNA has been observed to change in plants in response to biotic and abiotic stress factors. Little information has been reported on alterations in DNA methylation in orthodox tree seeds in response to storage conditions. In the current study, epigenetic integrity was analyzed in seeds of *Pyrus communis* L. in response to conventional and cryogenic storage. The results indicate that conventional storage under optimal conditions resulted in a significant increase in $m^5C$. In contrast, a decrease in $m^5C$ level after cryostorage at high water content (WC) was observed, not only in seeds but also in 3-month-old seedlings which were smaller than seedlings obtained from seeds cryostored at optimal WC. This shows that non-optimal cryostorage conditions increase epigenetic instability in seeds and seedlings. Optimal procedures for germplasm conservation are very important for germplasm banking since they have serious implications for the quality of stored collections. Maintaining epigenetic integrity during WC adjustment and optimal storage is a characteristic feature of orthodox seeds. The current results underline the importance of proper protocols and techniques for conventional storage and particularly cryopreservation as a method for conservation of true-to-type germplasm for long periods.

**Keywords:** conventional storage; cryopreservation; desiccation; DNA methylation; epigenetic integrity; liquid nitrogen; seed viability

## 1. Introduction

The protection of world's plant biodiversity has become critically important, particularly since according to International Union for Conservation of Nature (IUCN), 22% of all plant species are estimated to be under the threat of extinction [1]. One of the strategies of plant protection and maintaining biological diversity is ex situ conservation [2] which depends mostly on the storability of seeds [3] providing significant inter- and intrapopulational as well as cellular variability [4]. Since seeds are a plant's vector of moving through time and space to secure the next generation [4], the preservation of seeds in gene banks contributes to the prevention of the plant species extinction. Seed banking is the most economically efficient method for the long-term conservation of plant germplasm [5].

The most important factors affecting the viability of stored seeds are their water content (WC), storage temperature and the interaction between these two factors [6,7]. Orthodox seeds show a high level of tolerance to desiccation, which makes them easier to store for longer periods than recalcitrant (desiccation-sensitive) seeds. They down-regulate their metabolism, and have several cellular adaptations that protect them from desiccation-related injuries [8–10]. According to the recommendations of the Food and Agricultural

Organization (FAO) of the United Nations, orthodox seeds should be stored in gene banks at a water content (WC) of 0.03–0.07 g·g$^{-1}$ at $-18$ °C [11]. This recommended temperature, however, is not low enough to stop seed aging and prevent a decline in seed viability [12,13]. According to Colville and Prichard [1] standard conditions for *ex situ* seed conservation may not be sufficient for long-term preservation of a significant number of species and one of the promising ways to extend seed life span is temperature lowering [5]. Therefore, to keep plant germplasm safe for years or decades, cryopreservation is used when longer storage periods are required and used as a backup for standard collections [14–17]. The projected life span of seeds stored in liquid nitrogen (LN) vapors ($-135$ °C) could be 500 years, while seeds stored in LN ($-196$ °C) can last 3400 years [13]. This means that LN storage represents a 70-fold increase in storage time compared to conventional storage of orthodox seeds. Moreover, cryostorage is being utilized more frequently since it is very cost-effective and effectively prevents the infection of stored seeds by fungi and bacteria [18]. Cryopreservation is also important for the long-term storage of inherently short-lived orthodox and intermediate seeds as well as explants derived from recalcitrant seeds [13,19–22].

The successful preservation of biological material in gene banks requires that the conserved material maintains stable on physiological, biochemical, genetic (DNA sequence) and epigenetic (cytosine methylation, modifications of histones) levels. Epigenetic modifications play an important role in the regulation of gene expression, impacting the transcription of both coding and non-coding elements. In the process of seed development, CG- and CHG-context DNA methylation does not change significantly, however, in CHH-context DNA methylation increases primarily within transposons leading to dormancy, constituting a fail-safe mechanism to ensure that transposons remain silent and do not inactivate genes essential for seed development and germination [23,24]. As DNA methylation affects chromatin structure by inducing its condensation, it also modifies the interaction between transcription factors and DNA within the promoter region of genes [25,26]. However, it was shown that genes essential for seed development, including those encoding storage proteins, fatty acid biosynthesis enzymes and transcription factors, are located within regions hypomethylated throughout the plant life cycle [24]. Therefore, a unique, yet elusive, regulatory feature which occurs when a large number of seed genes are regulated in the absence of methylation changes within their promoters or gene bodies during development, is executed by an interaction between specific transcription factors and epigenetic events at the chromatin level [24]. Nevertheless, it was proposed that the influence of DNA methylation on gene expression may be primarily mediated by *trans* effects exerted by specific transcription factors and/or other regulatory proteins [27,28]. It has been estimated that up to 40% of the total cytosines in the genomes of angiosperms are methylated, mostly in heterochromatin regions [29,30].

Although methylated cytosine is considered to be a relatively stable epigenetic mark, changes in the level of m$^5$C have been observed in plants in response to both biotic and abiotic stress [31–34]. The ability of plants to acclimate and/or tolerate biotic and abiotic stress reduces the level of stress-related injury. Since stress-induced plant cell reprogramming involves changes in DNA structure and gene expression [26], all epigenetic modifications are considered to be significant drivers of genome flexibility. However, they may be the cause of undesirable germplasm variability. For instance, offspring of a cross between a hypomethylated and a normally methylated *Arabidopsis thaliana* plants induced novel epialleles and resulted in TE reactivation. Changes in DNA methylation may continue following whole-genome or gene duplication. Such genome–epigenome interactions shape genome evolution and phenotypes [30]. Nevertheless, they may be a threat to desired true-to-type germplasm preservation and plant regeneration when it is assumed that plants characterized by deviation in DNA methylation may have resulted from improper seed storage.

Therefore, even though plant genetic stability after germplasm preservation, including cryopreservation is under investigation, the effect of preservation techniques, including

conventional and cryostorage on epigenetic stability need further studies [35–37]. It is assumed that pre-storage procedures such as desiccation and cryoprotection as well as storage in LN itself, can affect m$^5$C (5-methylcytosine) content in the DNA of seeds and explants [38]. It was shown that in *Swietenia macrophylla* King seed germplasm changes in chromatin conformation after recovery from LN were observed, possibly due to changes in the methylation status of the DNA [39]. Significantly, although cryostorage of whole seeds is commonly used in many gene banks, there is insufficient information on the influence of this storage on their epigenetic integrity [40,41]. To address this issue, the present study was designed to investigate the epigenetic changes in seeds of *Pyrus communis* L. which is an important crop-related wild species.

It is considered that crop-related wild plant species are a rich source of genetic diversity and are potentially useful in plant breeding for the development of varieties with novel traits [42]. However, many crop wild relatives are poorly represented in gene banks and in other ex situ genetic resources collections [43,44]. Moreover, according to CWRnl [45] in pessimistic climate change scenarios the expected distribution area of *P. communis* in 2070 will be much narrower, as this species will not be present anymore in Italy, the Balkans region, Turkey, Czech Republic and Austria and will lose more than 90% of its natural distribution area in Spain and Poland, more than 75% in France and more than 50% in Germany. Therefore, the present study is focused on epigenetic stability measured by m$^5$C content in *P. communis* seeds subjected to changes in WC, cryogenic treatment and conventional storage in a refrigerator. Non-optimal liquid nitrogen (LN) storage conditions were implemented (high seed WC), to provide the answer on cryoinjuries affecting global DNA methylation stability and the potential to transfer these changes into regenerated plants. Moreover, it was uncertain whether it was possible to implement a mixed storage regime and treat orthodox seeds with LN after conventional storage (2 years) and obtain high germination and seedling emergence level. After a mixed storage regime, a global methylation analysis was then used to observe any potential deviation from standard conventional storage in refrigerator or cryostorage. The study also tried to answer whether maintaining epigenetic integrity is associated with the post-harvest physiology of seeds and if DNA methylation changes caused by cryopreservation of seeds can be observed in regenerated plants. This research aimed at providing a better understanding of the influence of seed banking procedures on the epigenetic integrity of seeds and regenerated plants, which is crucial for the protection of plant biodiversity.

## 2. Materials and Methods

### 2.1. Plant Material

Seeds of the common pear *(Pyrus communis* L.) collected in Łopuchówko, (52°35′57″ N, 17°05′58″ E) and Borkowice, (52°12′30″ N 16°47′05″ E), Poland, were manually extracted from mature fruits (green-yellow). After extraction, the WC of seeds was 0.4 g $H_2O$ g$^{-1}$ dry mass (g·g$^{-1}$). Long term storage of pear seeds was based on previously published methods [46,47]. After harvesting, pear seeds were dried on a laboratory bench at 20 °C to a WC of 0.08–0.09 g·g$^{-1}$ and were then placed in a drying box on blotting paper and desiccated for 5–14 days. To obtain a higher WC (0.21–0.41 g·g$^{-1}$), the seeds were sprayed several times with water until they reached a specified weight and were then left in tightly closed containers for 5–7 days at 3 °C. The required seed mass, corresponding to the measured WC, was calculated using a previously published formula [48]. The WC of seeds was assessed by drying them at 103 °C $\pm$ 2 °C for 18 h. Each analysis consisted of three replications of ten seeds each. Seeds stored for 24 h in LN were used in the study, as well as seeds conventionally stored at 3 °C for 24 months.

### 2.2. Conventional Storage and Cryostorage of Seeds

To assess the tolerance of seeds to ultra-low temperature, the seeds were placed in 1.8 mL cryovials, directly cooled in liquid nitrogen (LN), and then stored for 24 h. Samples were then thawed at 42 °C in a water bath for ten minutes prior to use. Conventionally

stored seeds of *P. communis* were kept at 3 °C in polypropylene bags for 24 months and then rewarmed to ambient temperature before use.

### 2.3. Stratification

Stratification of the dormant seeds was required before performing subsequent germination and seedling emergence tests. Seeds of *P. communis* were placed in plastic boxes in a moist mixture (1:1, *v/v*) of quartz sand (<1 mm fraction) and sieved peat with a pH 3.5–4.5. This substrate was kept moist. Seeds were mixed with the substrate (1:3, *v/v*), placed in 0.25 L plastic bottles and monitored for fungal infections and any indications of germination (i.e., seeds with a radical 2–3 mm long). The appearance of seeds with an emerging radical was used as an indicator that seeds had been released from dormancy. The substrate was watered weekly if required. The seeds were stratified for 16 weeks at 3 °C. At that time, no more than 5% of the seeds had developed a visible radicle. The seeds were then subjected to a germination test.

### 2.4. Germination and Seedling Emergence Test and Seedling Growth

Stratified seeds of *P. communis* were germinated in the dark, in a mixture of sand and peat identical to that used for stratification. Optimum thermal conditions for germination were ensured by subjecting the seeds to cyclically alternating temperatures of 3 °C/20 °C (16 h/8 h per day, respectively). A seed with a protruding radicle 3 mm long was considered to have germinated.

Seedling emergence tests were conducted in the same substrate used for stratification and germination tests. Stratified seeds were sown at a depth of 1 cm in plastic boxes containing the substrate and covered with a layer of sand. The seedling emergence test was conducted under the same thermal conditions as the germination test (3 °C/20 °C, 16 h/8 h per day). To ensure adequate moisture, the plastic boxes were covered with a transparent lid which was removed when the seedlings started to emerge. When seedlings were approximately 2–3 cm high, the boxes containing the seedlings were moved into an environmental chamber at 25 °C and provided light (16 h/8 h photoperiod at 60 $\mu$mol m$^{-2}$s$^{-1}$). The emergence test was conducted for 4–5 weeks.

For epigenetic analysis (determination of the m$^5$C content in leaf DNA), seedlings were grown for three months under a 16 h/8 h photoperiod at 77 $\mu$mol m$^{-2}$ s$^{-1}$ and at 20 °C. The shoot height (mm) of 3-month-old seedlings was measured from the root collar to the hoot apex of the germinated seedling. Each experiment was replicated four times and contained 25 seedlings per replicate.

### 2.5. DNA Isolation and Assessment of Global DNA Methylation Levels

A TLC-based method was used to measure changes in the level of DNA methylation in the whole genome. The total genomic DNA was extracted from five embryos (after removal of the seed coat) or leaves of seedlings with the Qiagen DNAeasy Plant Mini Kit$^{TM}$ (Qiagen, Hilden, Germany). Each bioreplicate was composed of five embryos and a total of five bioreplicates were examined in each treatment. Since each bioreplicate consisted of a mixture of five embryos or one-quarter of an apical part of three leaves from one seedling and each experimental treatment contained five replicates, the obtained data provided information on changes in m$^5$C in 25 seeds or 15 leaves for each treatment. Since the DNA methylation assessment was repeated five times, procedure-based variations in m$^5$C detection were minimized.

An analysis of the global content of m$^5$C in DNA of seeds was carried out and calculated as previously described [34]. Dried DNA (1 $\mu$g) was digested to completion with 0.001 U of spleen phosphodiesterase II and 0.02 U of microccocal nuclease in a 20 mM succinate buffer containing 10 mM CaCl$_2$ for 6 h at 37 °C. The resulting hydrolysate (0.3 $\mu$g) was labelled with 1.6 $\mu$Ci ($\gamma$-$^{32}$P) ATP (6000 Ci/mmol Hartmann Analytic, Braunschweig, Germany) and 1.5 U of T4 polynucleotide kinase in 10 mM bicine-NaOH buffer (pH 9.7) containing 10 mM MgCl$_2$, 10 mM DTT and 1 mM spermidine. After incubation for 30 min

at 37 °C, 0.03U of apyrase in 10 mM bicine-NaOH buffer was added and the mixture was incubated for 30 min. Subsequently, 0.2 µg of RNase P1 in 500 mM ammonium acetate buffer (pH 4.5) was used for 3′ phosphate cleavage. An analysis of ($\gamma$-$^{32}$P] m$^5$C was done by a 2-dimensional thin layer chromatography (2D TLC) on cellulose plates (Merck, Darmstadt, Germany) in isobutyric acid/NH$_4$OH/H$_2$O (66 mL/1 mL/17 mL, *v/v/v*), (first direction) and 0.1 M sodium phosphate pH 6.8/ammonium sulphate/n-propanol (100 mL/60 g/1.5 mL, *v/w/v*), (second direction). Radioactivity was measured with a Fluoro Image Analyzer FLA-5100 and Multi Gauge 3.0 Software.

For quantitative evaluation of the m$^5$C content, fluoroscopic image analysis accounted for the content of cytosine (C), thymine (T) and m$^5$C. All measurements of global DNA methylation levels were made in relation to pyrimidines C and T, because demethylation of m$^5$C to C can be accomplished by hydroxyl radical damage of the methyl group [49,50]. However, it is well known that the methylation of cytosine in DNA increases the rate of the hydrolytic deamination of m$^5$ resulting in the formation of T [51]. Therefore, T and C, as products of m$^5$C damage, are included in the calculation of R. The R ratio was calculated using the following formula:

$$R\,(\%) = I_{\text{m}}{}^5\text{C}/I_{\text{m}}{}^5\text{C} + I_\text{C} + I_\text{T} \times 100 \tag{1}$$

where *I* is the intensity of individual spots (clearly separated on a chromatogram), corresponding to the analyzed nucleotide.

### 2.6. Statistical Analysis

R statistical software (R Core Team 2020) was used for the statistical analysis of all the data. The fixed effects of storage conditions on germination and seedling emergence were evaluated separately using a generalized linear model (GZLM) with a binomial distribution. For the DNA methylation level and seedling height comparison, a linear regression model was used. DNA methylation and seedling height data were transformed prior to analysis according to Box Cox, and normality was assessed by Shapiro-Wilk test in the case of the linear regression models. A two-way analysis of variance (ANOVA) with the interaction between main effects (moisture content, and storage temperature), was used to analyze the significance of differences between means. LSD test (germination and seedling emergence) and Tukey test (DNA methylation and seedling height) were used for pair-wise comparisons, at a significance level of $p \leq 0.05$. Separate ANOVAs and post-hoc tests were performed for the analysis of germination, seedling emergence, seedling height and global DNA methylation. Significant differences between individual mean values of germination and emergence are indicated by different lower-case letters. Values labelled with the same letter were not significantly different at $p \leq 0.05$. The impact of each experimental treatment on seed germination and seedling emergence was assessed using four replications of 50. Data visualization was carried out in R with ggplot2 package.

## 3. Results

### 3.1. Germination, Seedling Emergence and DNA Methylation in P. communis Seeds after WC Adjustment and Cryostorage for 24 h

The germination of seeds at a WC of 0.05–0.35 g·g$^{-1}$ and stored in LN for 24 h varied from 88% to 100%. The germination percentage of these seeds did not differ significantly from seeds at the same range of WC not subjected to LN (-LN) but stored at 3 °C, except seeds at WC of 0.35 g·g$^{-1}$ (Table 1). Seeds with a WC of 0.41 g·g$^{-1}$ did not tolerate immersion in LN and, as a result, did not germinate after warming.

**Table 1.** Effect of storage at different levels of water content (WC) for 24 h in liquid nitrogen (LN) on *P. communis* seed germination and seedling emergence. Seeds were not conventionally stored prior to cryopreservation. Statistical analysis was conducted using the GLMZ and LSD test. The values marked with the same letter are not significantly different at $p < 0.05$.

| Seed Water Content, (g·g$^{-1}$) | LN Treatment | Germination (%) | Seedling Emergence (%) |
|---|---|---|---|
| 0.05 | NO | 98 (a) | 99 (a) |
| 0.05 | YES | 99 (a) | 99 (a) |
| 0.09 | NO | 99 (a) | 99 (a) |
| 0.09 | YES | 100 (a) | 99 (a) |
| 0.21 | NO | 97 (a) | 90 (a) |
| 0.21 | YES | 97 (a) | 100 (a) |
| 0.25 | NO | 100 (a) | 99 (a) |
| 0.25 | YES | 97 (a) | 100 (a) |
| 0.35 | NO | 95 (a) | 100 (a) |
| 0.35 | YES | 88 (b) | 77 (b) |
| 0.41 | NO | 97 (a) | 99 (a) |
| 0.41 | YES | 0 (c) | 0 (c) |

Seedling emergence for seeds in the WC range of 0.05–0.25 g·g$^{-1}$ and stored in LN was 96%–100%. This level of emergence did not differ significantly from that of seeds in the same range of WC but which had not been subjected to cryostorage (-LN). In contrast, seedling emergence of seeds immersed in LN for 24 h at a higher WC (0.35 g·g$^{-1}$) was significantly lower (77%) than for seeds at the same WC but stored at 3 °C (Table 1). No seedlings were observed in seeds cooled in LN at the highest WC (0.41 g·g$^{-1}$).

The level of global DNA methylation in seeds at a WC of 0.09 g·g$^{-1}$, not stored in LN, was R = 3.08% (Figure 1). The differences in the m$^5$C level between seeds immersed for 24 h in LN and those not subjected to LN were not statistically significant when WC was in the range of 0.05–0.21 g·g$^{-1}$. Cryostorage of seeds for 24 h at a higher WCs caused a significant decrease in the level of global DNA methylation compared to seeds that were not stored in LN by R = 1.11%, 1.21% and 1.54% for 0.25, 0.35 and 0.41 g·g$^{-1}$, respectively.

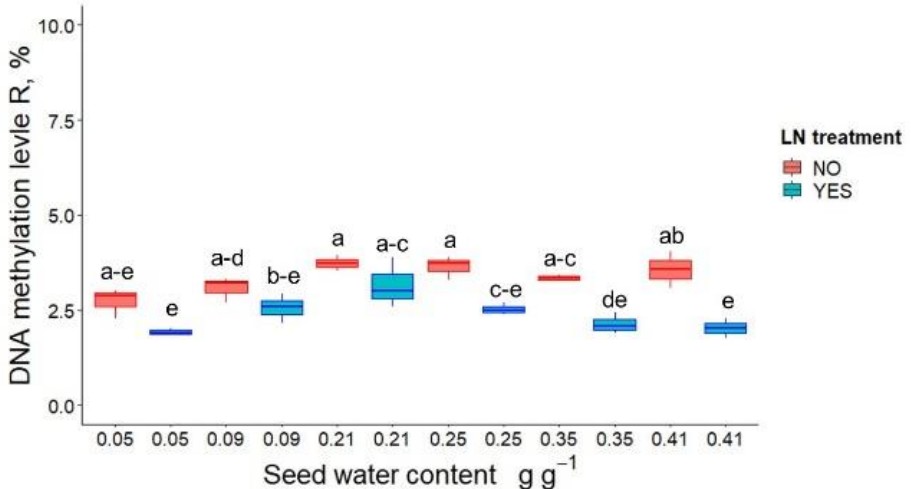

**Figure 1.** Effect of storage at different levels of WC in LN for 24 h on the global DNA methylation of *P. communis* seeds. Seeds were not conventionally stored prior to cryopreservation. Statistical analysis was conducted using ANOVA and a Tukey test. Values marked with the same letter are not significantly different at $p < 0.05$. Lines denote the median, boxes the 25th to 75th percentile, while whiskers are the 5th to 95th percentiles.

### 3.2. Germination, Seedling Emergence and DNA Methylation in P. communis Seeds Conventionally Stored for 2 Years Prior to Cryostorage

Prior to cryostorage, seeds were conventionally stored in a refrigerator at 3 °C for 2 years at a WC of 0.08 $g \cdot g^{-1}$ and a portion of these seeds were then desiccated to a lower WC level of 0.05 $g \cdot g^{-1}$. The seeds were immersed in LN for 24 h and then warmed. The level of germination of these seeds was 98%–100%. These levels of germination were similar to those obtained for seeds conventionally stored for two years, that had been desiccated to the same range of WC, but not subjected to immersion in LN (Figure 2A). Germination of two-year-old seeds whose WC had been raised to 0.35 $g \cdot g^{-1}$ prior to cryostorage, decreased significantly to 35%, whereas the level of germination in seeds at the same WC not immersed in LN remained high (98%) (Figure 2A).

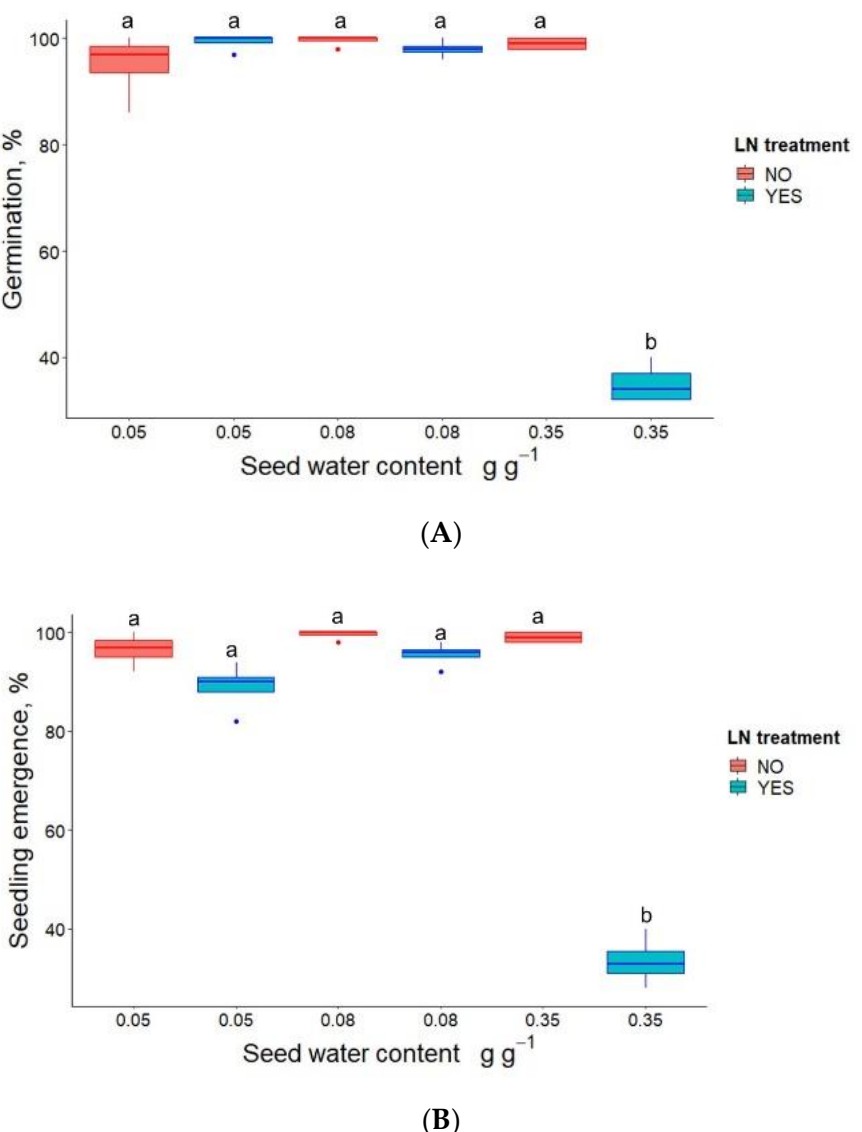

(**A**)

(**B**)

**Figure 2.** Effect of storage at different levels of WC in LN for 24 h on *P. communis* seed germination (**A**) and seedling emergence (**B**). Seeds were stored for two years at 3 °C at a WC of 0.08 $g \cdot g^{-1}$ prior to cryostorage. Statistical analysis was conducted using GLMZ and LSD test. The values marked with the same letter are not significantly different at $p < 0.05$. Lines denote the median, boxes the 25th to 75th percentile, while whiskers are the 5th to 95th percentiles.

Seedling emergence from seeds not stored in LN (-LN) was 96%–99%, with the highest value at a WC of 0.08 g·g$^{-1}$. Seeds cryostored at a WC of 0.05 or 0.08 g·g$^{-1}$ exhibited 89% and 96% seedling emergence, respectively (Figure 2B). Seeds with high WC (0.35 g·g$^{-1}$) had significantly lower levels of seedling emergence (34%) after immersion in LN compared to seedling emergence (99%) from seeds not subjected to cryostorage (Figure 2B).

The level of a global DNA methylation in seeds after 2-year storage at 3 °C, prior to cryostorage, was R = 5.75% (0.08 g·g$^{-1}$ WC). The m$^5$C level in seeds at a WC of 0.05 and 0.08 g·g$^{-1}$ that had been treated with LN for 24 h did not differ significantly from the level of DNA methylation observed in non-cryostored seeds at the same WC. DNA in seeds at higher (0.35 g·g$^{-1}$) WC and immersed in LN for 24 h exhibited a large decrease in the global level of methylation level compared to seeds not subjected to such treatment (6.75% vs. 4.19%, respectively), (Figure 3).

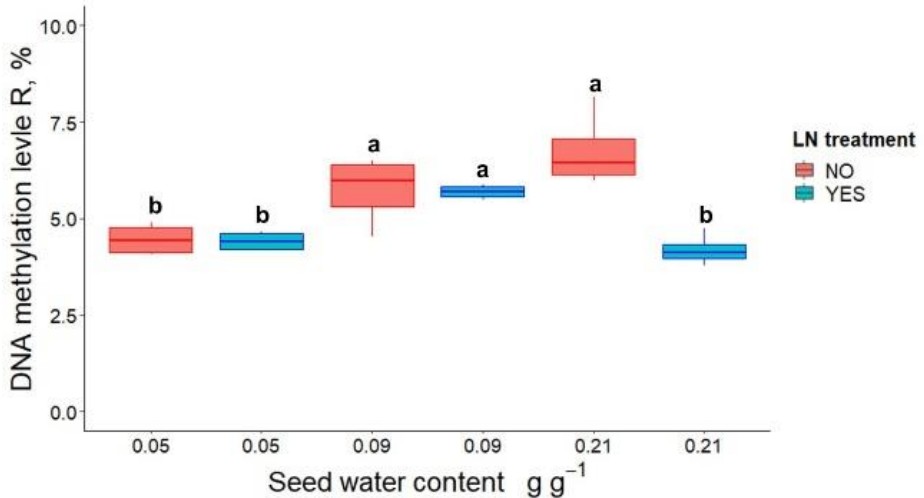

**Figure 3.** Effect of storage at different levels of WC in LN for 24 h on the global DNA methylation of *P. communis* seeds. Seeds were stored for two years at 3 °C at a WC of 0.08 g·g$^{-1}$ prior to cryostorage. Statistical analysis was conducted using ANOVA and Tukey tests. Values marked with the same letter are not significantly different at *p* < 0.05. Lines denote the median, boxes the 25th to 75th percentile, while whiskers are the 5th to 95th percentiles.

### 3.3. DNA Methylation and the Height of Seedlings Derived from Cryostored Seeds

The global DNA methylation level in seedlings obtained from seeds cryostored in the range of WC 0.05–0.21 g·g$^{-1}$ did not differ significantly from control seedlings. When seeds at a WC of 0.25 and 0.35 g·g$^{-1}$ were compared, the m$^5$C level in seedlings obtained from these seeds significantly decreased by 2.32% and 3.98%, respectively (Figure 4A). The height of seedlings regenerated from seeds desiccated to 0.09 g·g$^{-1}$ did not differ when compared to seedlings regenerated from cryostored seeds, while for seeds at high WC, the decrease in seedling height was significant and equaled 22.5 mm (Figure 4B).

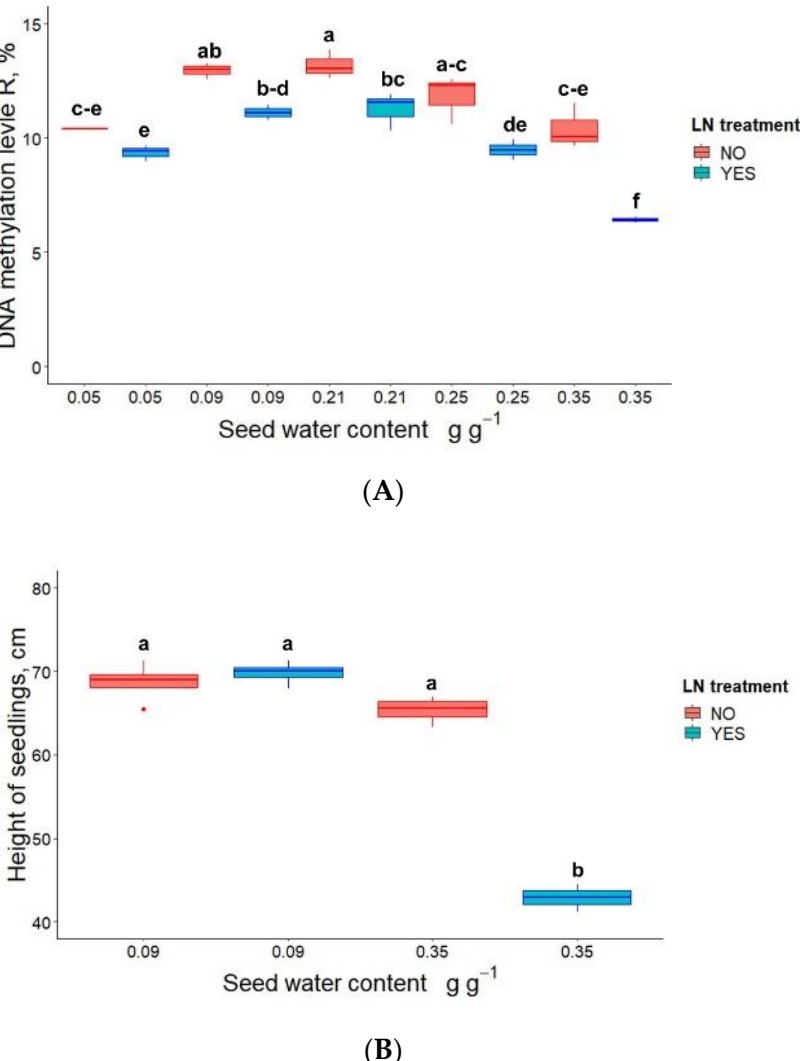

**Figure 4.** Effect of storage at different levels of WC in LN for 24 h on the global DNA methylation of *P. communis* seedlings (**A**) or height of seedlings (**B**). Statistical analysis was conducted using ANOVA and Tukey tests. Values marked with the same letter are not significantly different at $p < 0.05$. Lines denote the median, boxes the 25th to 75th percentile, while whiskers are the 5th to 95th percentiles.

## 4. Discussion

Orthodox seeds are defined by their ability to maintain viability for extended periods of time after they are significantly desiccated to a 3%–7% moisture content (MC). In such conditions, cytoplasm solidifies, forming a glass, and chemical reactions slow down [52,53]. Because of desiccation-tolerant characteristics, the cryostorage of orthodox seeds is relatively straightforward and does not require complicated, multistep pre-storage treatments as in the case of recalcitrant seeds to prevent ice nucleation and cryoinjuries [6,54]. However, even though the amount of genetic diversity preserved by conventional and cryostorage is comparable [2], there is still insufficient knowledge about the impact of cryopreservation on molecular processes in orthodox seeds. One of the reasons is that, for many years, the majority of studies focused only on the effect of cryostorage on germinability. Significantly, there are only a few studies showing the effect of cryostorage on DNA methylation status of orthodox seeds [39] or plants regenerated from these seeds [40]. Secondly, when the effect of cryostorage on epigenetic stability is analyzed, some discrepancies can be noted in the literature. While changes in global DNA methylation after cryopreservation have been previously reported, most of these studies were conducted, not using seeds but explants

regenerated in vitro. Therefore, observed changes in DNA methylation might be due to an adaptive response to the stress associated within vitro culture and treatment prior to cryostorage rather than a direct response to cooling in LN [20,55–60].

Changes in the level of methylation were also considered to be due to the effect of a cold acclimating treatment (hardening at $-1$ °C) prior to cryopreservation or in vitro culture following cryostorage rather than storage in LN *per se* [41,58]. For this reason, it was decided to take an advantage of the orthodox characteristics and investigate the direct effect of LN storage on seed epigenetic integrity concerning seed WC status since the procedure of cryogenic immersion of orthodox seeds is devoid of cryoprotectants (chemical treatment) and explant isolation (mechanical injuries), possibly affecting the epigenetic response. Importantly, in addition to seed germinability data, seedling height and epigenetic status was also provided.

As shown in Figures 1 and 2 at WC $\leq 0.25$ g·g$^{-1}$ there was no statistical difference in germination and seedling emergence between cryostored and control seeds. As also reported by Reed et al. 2001 [5], a high germinability of *P. communis* seeds was observed after cryostorage of seeds at MC of 5% and 10%. However, when epigenetic integrity was analyzed, a decrease in m$^5$C level was detected at WC of $\geq 0.25$ g·g$^{-1}$. A further analysis of epigenetic status in regenerated seedlings indicated a statistically significant difference even at WC of $\geq 0.21$ g·g$^{-1}$. The difference in seedlings height was also detected after three months between seedlings obtained from seeds stored in LN at optimal and at WC of 0.35 g·g$^{-1}$. Therefore, these results show that cryostorage of orthodox seeds is not damaging for their epigenetic integrity and allows true-to-type plants to be regenerated unless tissues are subjected to LN at a non-optimal, high water content that exceeds the amount of water molecules limiting freezing. Villalobos et al. [61] showed that even though young 7–14-day seedlings of sorghum derived from cryostored and control seeds revealed biochemical and morphological difference, these differences became negligible after 110 days of growth. In the current study, a difference was still observed in m$^5$C level in DNA isolated from 3-month-old seedlings as well as a decrease in the height of seedlings. This means that cryo-related epigenetic instabilities can be transferred through mitotic cell divisions into regenerated plants. Previously, Lu et al. [41] found that there was no significant effect on DNA methylation status or phenotypic traits in seedlings derived from orthodox *Secale cereale* L. seeds, although they were cryostored only at optimal WC.

It is important to note that lower level of m$^5$C was observed in seedlings obtained from seeds cryopreserved at highest WC, but also in seedlings derived from cryostored and non-cryostored seeds at the lowest WC of 0.05 g·g$^{-1}$. This indicates that over-drying of seeds should be avoided before cryopreservation, as it can affect the epigenetic landscape. Even though several previous reports suggested that seed cryopreservation can affect biochemical processes in regenerated seedlings [62], because the effect of cryostorage at different WC was not taken into consideration, it was not assessed whether cryostorage of seeds at low WC could have any effect. The current results are in concordance with Walters and Engels [63], who reported that drying to extremely low WC may decrease seed viability. They indicated that protocols for cryopreservation of seeds in gene banks should be preceded by very detailed studies, including those on regenerated seedlings. The current results also support this view, as changes in epigenetic integrity were not seen at the moment of seed germination, although they were statistically significant in seedlings obtained from seeds at WC different than 0.09 g·g$^{-1}$. Therefore, valuable orthodox seed material should be desiccated to optimal WC of 0.09–0.10 g·g$^{-1}$ before cryopreservation, as this seems to be a safe WC for maintaining epigenetic stability.

However, the damaging effect of cryostorage at high seed WC has been already described in detail [2,22,64,65]. It was shown that the WC of 0.2–0.3 g·g$^{-1}$ is sufficient to allow ice formation at $-10$ to $-40$ °C in relevant time frames [66]. Consequently, thermal-stress induced fractures of biological material may cause serious damage to stored samples [64]. However, the idea that ice crystals cause immediate physical damage to cells was challenged by Wesley-Smith et al. [67,68], as it was shown that embryonic axes from

*Acer saccharinum* L. remained intact after plunge-cooling in LN and fast thawing (1 min. at 40 °C). To date, there is no evidence of immediate mechanical post cooling-thawing fragmentation of DNA in plant cells that would possibly facilitate removal of the $m^5C$ or methyl group or affect the detection of $m^5C$ in cryostored material. However, DNA and chromatin fragmentation in cryopreserved cells is still under debate, at least in animal research models, as contradictory results have appeared. Thus, even if the results suggest LN-dependent DNA fragmentation leading to genome instability and defects in epigenome maintenance, the data were acquired 0.5 h or later after thawing [69], so possibly not presenting the direct and immediate fracturing of DNA, but rather the progression of intracellular processes related to damage signaling. In the current experimental set-up, $m^5C$ evaluations were initiated immediately after the rewarming of seeds. Thus, the changes in the level of DNA methylation were not observed to be relate to the progressive deterioration of seeds (finally manifested as lack of germination and seedling emergence) but rather all changes in $m^5C$ resulted from the response of seeds to cooling and thawing. Consequently, the rewarming of seeds was recognized as the moment with the greatest potential for causing disruption in cell structure and affecting epigenetic integrity as seeds were rewarmed by placing in a water bath at 42 °C for 10 min, which is a slower rate than cooling by immersion in LN [70] and potentially the reason for ROS supra-physiological levels [71]. It was already described that cellular membranes and macromolecules (DNA and proteins) are targets for ROS-driven cryoinjuries [18]. Therefore, the decrease in $m^5C$ at high WC of $\geq 0.21$ g·g$^{-1}$ may result from different processes, including enzymatic activity (demethylases) and/or ROS oxidation of $m^5C$ and removal of damaged bases that may occur while seeds are being thawed. ROS can directly affect methylation due to their ability to modify 5-methylcytosine ($m^5C$) to 5-hydroxymethylcytosine ($hm^5C$), 5-formylcytosine ($f^5C$), 5-carboxylcytosine ($ca^5C$) and other derivatives. Therefore, the presence of high levels of ROS can result in elevation of oxidized forms of $m^5C$ [49,50]. Indeed, environmental stresses were demonstrated to cause a significant alteration in the level of oxidized forms of $m^5C$ that can be subsequently removed by the active removal of both $m^5C$ or its oxidized derivatives [49]. Nevertheless, further investigations in this area are needed to better understand the processes occurring during cooling by immersion in the LN/thawing procedure. Moreover, the question of possible ROS-driven changes in global $m^5C$ appearing while storage in LN remains. Since most biological methods do not work in a low-temperature environment, the mechanism and details of the depression of cellular activity in the frozen state remain largely uncharacterized. However, it was demonstrated that the temperature dependence of native redox reaction rates can be described by the thermal activation law with an apparent energy of 32.5 kJ/mol, showing that the redox reaction rate is ~$10^{15}$ times slower at LN temperature than at room temperature [72]. Moreover, it is widely agreed that during storage at LN, metabolic and most physical processes are stopped [39]. Therefore, we consider the thawing process as a cause of post-cryostorage epigenetic instabilities that may result both from ROS and enzymatic activity.

It was also tested whether the successful cryostorage of seeds that had been already pre-stored could be achieved. This study is particularly important in terms of the prolongation of germplasm longevity and is potentially useful for seed banks and initiation of LN back-up collections of orthodox seeds that have been already preserved under conventional conditions. Indeed, ex situ preservation of germplasm is recognized to play a significant complementary work along with in situ preservation [21,73]. However, storage under conventional conditions in refrigerator results in a slow, but gradual, loss in seed viability and requires costly investment in equipment, maintenance and rejuvenation of the collections. These costs may be drastically reduced by LN storage [21,39]. Therefore, the *P. communis* seeds were kept at 3 °C for two years. Germination and seedling emergence of these seeds remained unchanged among the entire tested WC range. This result is contrary to Walters et al. 2004, who showed that lettuce seeds (orthodox) revealed lower viability after pre-storage at 5 °C after subsequent cryostorage. Nevertheless, identically as shown previously for seeds at ~0.9 g·g$^{-1}$ after 12 months of storage [33], the DNA

methylation level was higher in seeds stored for 24 months at 3 °C compared to non-stored seeds (Figures 2 and 4). Since cell division and DNA replication do not take place in dormant seeds, it can be suggested that RNA-directed DNA methylation (RdDM) and DOMAINS REEARRANGED METYLTRANSFERASE 2 (DRM2) are involved in *de novo* hypermethylation of genomic DNA. It was already shown that dry seeds store substantial levels of RNA transcripts for components of RdDM, including DRM2 and that RdDM is active at 4 °C. However, opposite to our findings, it was also suggested that upon complete desiccation DNA methylation may be halted. Nevertheless, long term storage was not taken into consideration [22].

To the contrary of the presented research, when embryonic axes of *Quercus robur* L. (recalcitrant) were analyzed, a significant correlation was shown between aging, manifested as a decrease in seed germination and seedling emergence, and m$^5$C decline [74]. Significantly, it was also recently reported that epigenetic changes were detected with the MSAP technique upon accelerated aging conditions in orthodox seeds of *Secale cereale* L. and *Mentha aquatica* L., even before a major loss of viability was observed [75,76]. Therefore, it can be summarized that an increase in global DNA methylation level would block DNA damage and therefore may contribute to desiccation tolerance. It may be also associated with reinforcement of chromatin condensation, prevention of unfavorable gene expression and TE activation and finally with prolonged seed longevity.

This is the first time that such comparison between epigenetic integrity of recalcitrant and orthodox seeds has been made, as there are no other results comparing the differences in m$^5$C change in seeds of different post-harvest physiology after conventional storage without the implementation of accelerated aging protocols. The observed increase in global levels of DNA methylation during conventional storage could be connected with the slowing of biological processes in seeds and may represent an adaptive trait of orthodox seeds in acquiring a quiescence state. Indeed, the maturation of seeds concomitant with their WC decrease has already been connected to the repression of the seed genome in their parent plants [77]. A hypothetical mechanism for this phenomenon would involve enzymatic regulation and an increase in m$^5$C amount to be driven by DNA methyltransferases. All of this supports the concept of DNA methylation as a plausible molecular marker of the viability of seeds with varied post-harvest physiology [74,78].

## 5. Conclusions

The genetic and consequently epigenetic integrity of cryo-banked germplasm has important implications for long-term stability and viability of seeds. No previous studies have reported results on the epigenetic integrity of orthodox (desiccation-tolerant) seeds subjected to conventional and LN storage. The current study proved that cryostorage can be a safe method of orthodox seeds conservation since no epigenetic instability was observed at optimal seed WC. This statement is also valid for seeds cryostored immediately after WC adjustment as well as for seeds pre-stored for two years at optimal WC. However, the global DNA methylation level was observed to decline when seeds were stored in LN at non-optimal WC. Significantly, epigenetic instabilities were transferred to seedlings and maintained for at least three months. Seedlings showing a decrease in m$^5$C level were also smaller than seedlings derived from control seeds. These data clearly show that cryopreservation of orthodox seeds in wide, so-called "safe range of MC", based only on germination or seedling emergence data without confirmation on molecular stability, may lead to storage of not true-to-type germplasm. Finally, an increasing global DNA methylation level may be connected with orthodox characteristics and maintaining viability during storage. Assessment of m$^5$C may be considered as a viability marker of seeds and a supporting criterion for seed classification into proper post-harvest physiology groups.

**Author Contributions:** Conceptualization, M.M., B.P.P.-M. and P.C.; methodology, M.M., M.Z.N.-B., P.C.; validation, M.M.; formal analysis, M.M.; investigation, M.M., M.Z.N.-B.; resources, M.M., M.Z.N.-B.; data curation, M.M., and B.P.P.-M.; writing—original draft, B.P.P.-M., M.M.; writing—review and editing, P.C., M.Z.N.-B., J.B.; visualization, M.M.; supervision, M.M., M.Z.N.-B., J.B., and

P.C.; project administration, M.M., P.C.; funding acquisition, M.M., P.C. All authors have read and agreed to the published version of the manuscript.

**Funding:** This research was partially funded by National Science Center, Poland, grant number N309 072036; and grant number 2011/01/B/NZ9/02864.

**Data Availability Statement:** The data presented in this study are available on request from the corresponding author.

**Acknowledgments:** We greatly appreciate the technical support provided by Elżbieta Drzewiecka-Pieniężna.

**Conflicts of Interest:** The authors declare no conflict of interest.

## Abbreviations

| | |
|---|---|
| 2D TLC | 2-dimensional thin-layer chromatography |
| C | cytosine |
| CWRnl | Crop Wild Relatives (CWR) in the Netherlands |
| HPLC | high-performance liquid chromatography |
| LN | liquid nitrogen |
| $m^5C$ | 5-methylcytosine in DNA |
| R | DNA methylation level expressed in % according to the formula provided in the manuscript |
| T | thymine |
| WC | water content |

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
