# Peer review of "Epigenetic Integrity of Orthodox Seeds Stored under Conventional and Cryogenic Conditions"

_forests, doi:10.3390/f12030288_

Round 1

Reviewer 1 Report

Dear authors,

I would like to thank You for Your manuscript entiteld iEpigenetic integrity of orthodox seeds stored under conven-tional and
cryogenic conditions.
It is very important to collect old varieties. 

It would be more vunerable if You did the experiment with common pear from other locations, not only Lopuchowko and Borkowice. It would be interesting for future experiment.

There are several changes:

line 48 - reference 10, not 11

line 49 - 11-13, not 12-13 (check the whole reference list)

line 157, 167 - latin  names italic

For the Figure 1 A and B - do not start at 0, reather from 50. The data will be more visiable

Do You think that only box plot diagrams are good way to show results? Try to show the data with different figures and/or tables. 

line 346 - Reed et al. - number of reference

line 524 - Seeds

line 524- ex situ - italic

line 598 - date 25 11 2020

Author Response

Thank you very much for your comments. We adapted the manuscript to all sugestions.

Response to Reviewer
line 48 - reference 10, not 11 (done)

lline 49 - 11-13, not 12-13 (check the whole reference list) (done)

line 157, 167 - latin  names italic (done)

For the Figure 1 A and B - do not start at 0, reather from 50. The data will be more visiable - done as we changed Figure 1 into Table 1

Do You think that only box plot diagrams are good way to show results? Try to show the data with different figures and/or tables. -done as we changed Figure 1 into Table 1

line 346 - Reed et al. - number of reference (done)

line 524 - Seeds (done)

line 524- ex situ - italic (done)

line 598 - date 25 11 2020 (done)

Reviewer 2 Report

This manuscript presents “Epigenetic integrity of orthodox seeds stored under conventional and cryogenic conditions. In my opinion, the study is well executed and the manuscript is fairly written. My suggestion is author should take care in writing the botanical name particularly the full name of species in italic letters and the use of abbreviated terms. I would suggest the presentation of numerical data and statistical output in a table as well. Some minor comments are made in the attached pdf file.

Author Response

Thank you very much for your comments. We adapted the manuscript to all changes

Convn-tional changed to concentional

Line 19 WC changed to Water content
Line 92 A. thaliana changed to Arabidopsis thaliana
Line 121 LN changet to liquid nitrogen
Line 167 P. communis changes to italics P. communis
Line 277 Figure numbers was checked and corrected
Line 384 Sntence "However, the idea that ice crystals cause immediate physical damage to cells was chal-
lenged by Wesley-Smith et al. [67-68], as embryonic axes Acer saccharinum L. remained
intact after plunge-cooling and fast thawing (1min. at 40°C)."
changed to
However, the idea that ice crystals cause immediate physical damage to cells was challenged by Wesley-Smith et al. [67-68],
as it was shown that embryonic axes from Acer saccharinum L. remained intact after plunge-cooling in LN and fast thawing (1min. at 40°C).